# A Non-Contrastive Learning Framework for Sequential Recommendation with Preference-Preserving Profile Generation

**Huimin Zeng[1]\*, Xiaojie Wang[2]†, Anoop Jain[2], Zhicheng Dou[3], Dong Wang[1]**
[1]University of Illinois Urbana-Champaign [2]Amazon [3]Renmin University of China
```
{huiminz3, dwang24}@illinois.edu,
{xiojie, anoopjai}@amazon.com,
dou@ruc.edu.cn
```

## Abstract

Contrastive Learning (CL) proves to be effective for learning generalizable user representations in Sequential Recommendation (SR), but it suffers from high computational costs due to its reliance on negative samples. To overcome this limitation, we propose the first Non-Contrastive Learning (NCL) framework for SR, which eliminates the computational overhead of identifying and generating negative samples. However, without negative samples, it is challenging to learn uniform representations from only positive samples, which is prone to representation collapse. Furthermore, the alignment of the learned representations may be substantially compromised because existing ad-hoc augmentations can produce positive samples that have inconsistent user preferences. To tackle these challenges, we design a novel preference-preserving profile generation method to produce high-quality positive samples for non-contrastive training. Inspired by differential privacy, our approach creates augmented user profiles that exhibit high diversity while provably retaining consistent user preferences. With larger diversity and consistency of the positive samples, our NCL framework significantly enhances the alignment and uniformity of the learned representations, which contributes to better generalization. The experimental results on various benchmark datasets and model architectures demonstrate the effectiveness of the proposed method. Finally, our investigations reveal that both uniformity and alignment play a vital role in improving generalization for SR. Interestingly, in our data-sparse setting, alignment is usually more important than uniformity.

## 1 Introduction

Contrastive Learning (CL) proves to be an effective approach to learn generalizable user representations for sequential recommendation (SR) (Wang et al., 2022b; Liu et al., 2021; Xie et al., 2022). A canonical CL-based recommendation framework usually employs positive samples (e.g., augmented user profiles or items) to promote representation alignment and negative samples (e.g., profiles of other users or irrelevant items) for representation uniformity (Wang & Isola, 2020; Zhuo et al., 2023). With enhanced alignment and uniformity, CL-based methods achieve state-of-the-art performance.

However, CL-based methods inevitably suffer from high computational costs, because they heavily rely on negative samples to avoid representation collapse during training (Grill et al., 2020; Chen & He, 2021). For example, CL-based methods either require large batch sizes during training or use sophisticated retrieval strategies to identify hard negative samples (Robinson et al., 2020), leading to challenges on hardware memories and computational efficiency (Kulatilleke et al., 2023).

In a sharp comparison, Non-Contrastive Learning (NCL) emerges as a promising solution that avoids using negative samples (Zhuo et al., 2023). Unlike CL using negative samples for uniformity, NCL

---

\*Work done while interning at Amazon.
†Corresponding author.

is capable of learning uniform representations using only positive samples. Therefore, in this work, we aim to learn uniformity using only positive samples for SR. The rationale behind NCL is: such uniformity can best represent the information of individual data points, because a probability distribution that best represents the current state of knowledge about a system is the one with the largest entropy according to information theory (Jaynes, 1957; Guiasu & Shenitzer, 1985). Nevertheless, despite the success of NCL for Computer Vision (Grill et al., 2020; Zbontar et al., 2021) and Natural Language Processing (Cho et al., 2022; Zhou et al., 2023), NCL for recommendation is unexplored. Furthermore, learning generalizable user representations from recommendation data through NCL is much more complex and difficult than learning representations for images and texts.

The key challenge stems from obtaining high-quality positive user profile pairs to enable non-contrastive training. On one hand, within an NCL framework, the absence of negative samples makes it more challenging for the recommender to learn uniform representations from the sparse recommendation data than from dense CV or NLP data. This is because the sparsity of the recommendation data increases the risk of representation collapse Guo et al. (2023). In recommendation, the collapsed user representations are mapped nearby in the latent space, even if they represent drastically different user preferences. As a consequence, such collapsed representation lacks expressiveness to capture unique preferences of different users, making it difficult for the recommender to generate meaningful recommendations. On the other hand, the representation alignment may be substantially compromised as well, due to the inconsistent user preferences encoded in the positive samples. We note that existing operations are too ad-hoc to generate consistent positive user profile pairs. For instance, data augmentations are primary methods to generate positive samples, but they are usually random (e.g., random crop, random swap, or random mask (Xie et al., 2022)) or heuristic (correlation-based substitution or insertion (Liu et al., 2021)). Obviously, such ad-hoc operations may fail to preserve the genuine user preferences in the positive samples. These inconsistencies confuse the model, when the model is trained to align profiles that reflect different, even conflicting preferences for the same user. As a consequence, the representation alignment is substantially compromised, and the model fails to capture the true user preferences, degrading the performance.

Therefore, targeting the limitations of CL-based SR methods and NCL methods, we propose the first **N**on-**C**ontrastive **L**earning framework for **S**equential **R**ecommendation powered by preference-preserving user profile generation: **NCL-SR**. The proposed NCL-SR eliminates the computational overhead of identifying and generating negative samples in CL. Furthermore, we design a novel preference-preserving profile generation to address the representation collapse issue and preference inconsistency within the training data. Inspired by Differential Privacy (DP), our approach creates augmented user profiles that retain consistent user preferences with provable guarantees. To address representation collapse, our approach generates highly diversified positive samples with controllable randomness. The diversity of positive samples helps the model explore distinct and larger portions of the representation space, which effectively reduces the risk of representation collapse. In addition, the preference-preserving design of our approach also addresses the inconsistent preference issue within the positive samples by preserving the genuine user preferences within the generated positive samples. Finally, note that the DP augmentation serves as the key pre-requisite to promote both alignment and uniformity under our NCL framework. With the generated augmented user profiles, the proposed framework computes a non-contrastive alignment loss and a non-contrastive uniformity loss from matrix information theory (Zhang et al., 2023), to improve the generalization of the learned user representations. Overall, our contribution can be summarized as follows:

- We design and present the very first self-supervised non-contrastive learning framework for sequential recommendation: NCL-SR. NCL-SR learns generalizable and robust user representations for sequential recommendation.

- We present a novel data augmentation operation with theoretical guarantees to enable non-contrastive training. Unlike existing ad-hoc augmentation operations, our approach is guaranteed to preserve user preferences in the augmented user profiles, addressing the representation collapse issue and preference inconsistency issue within the training data.

- We conduct extensive experiments encompassing diverse datasets. Our experimental results suggest NCL-SR consistently exhibits superior performance over state-of-the-art SR models and CL-based SR methods. Our investigations reveal that uniformity and alignment of the representations play a vital role in SR. Interestingly, we observe that alignment may be more important than uniformity.

## 2 RELATED WORK

**Self-Supervised Learning for Sequential Recommendation.** There are three types of self-supervised learning schemes for sequential recommendation: generative, adversarial, and contrastive (Ren et al., 2024). Generative SR models primarily focus on generating sequence item data as recommendations (Ye et al., 2023; Wu et al., 2023; Wang et al., 2022d). An exemplar generative SR model is BERT4Rec (Sun et al., 2019), where the recommender is trained to generate and reconstruct the masked items within user histories. In comparison, adversarial SR models (Ni et al., 2023; Lv et al., 2021) are mainly GAN(Goodfellow et al., 2020)-like in the sense that they usually train discriminators that distinguish between real and generated item sequences, while optimizing generators to produce realistic recommendations. Compared to generative and adversarial SR models, contrastive SR models comprise the majority of self-supervised SR methods and are state-of-the-art. Existing CL-based SR methods focus on designing sophisticated rules and augmentation operations to construct contrastive pairs. For instance, CLS4Rec (Xie et al., 2022) adopts three randomized operations to generate augmented views of user history. Based on CLS4Rec, CoSeRec (Liu et al., 2021) further proposes two informative augmentations: correlation-based item substitution and insertion. Liu et al. (2021) present EC4Rec, an explanation-guided CL framework that identifies positive/negative items using training gradients. In comparison, DUORec (Qiu et al., 2022) showcases a supervised contrastive learning framework with model-level dropout augmentations to further improve representation learning. In (Zheng et al., 2022), aiming at the Cross-Domain Sequential Recommendation challenge, the authors propose a novel model via dual dynamic graph modeling and hybrid metric training with contrastive learning integration. In (Su et al., 2023), a novel hierarchy-aware dual clustering graph network (HADCG) model is designed to explore the inherent hierarchy structures from both item popularity and collaborations. With an information regularizer for intra-session clustering and contrastive training for inter-session clustering, HADCG achieves substantial performance improvements in the session-based recommendation. Finally, Shi et al. (2024) present self contrastive learning (SCL) to enhance the uniformity of the learned item representations. However, CL-based methods usually pose challenges to hardware memories and computational efficiency, because they either require large batch sizes or sophisticated retrieval strategies to identify hard negatives (Robinson et al., 2020). Furthermore, augmentation operations of CL-based methods may also introduce preference inconsistency in the positive samples, because they may fail to preserve user preferences. This makes CL-based methods sensitive to data augmentation.

**Non-Contrastive Self-supervised Learning.** Recently, there is an increasing interest in Non-Contrastive Learning (NCL) for its capability of learning robust representations without negative samples. Unlike CL which pursues both alignment and uniformity in representation learning, NCL tends to focus more on learning uniform data representations. For instance, BYOL (Grill et al., 2020) and SimSiam (Chen & He, 2021) introduce asymmetry in the network architecture and parameter updates to avoid representation collapse. Barlow Twins (Zbontar et al., 2021) enhances uniformity by making the cross-correlation matrix of two twin representations as close to the identity matrix as possible. Liu et al. (2022) present a theoretical framework based on the maximum entropy encoding principle from information theory, encompassing all previous NCL losses into a generalized uniformity loss. Finally, harmonizing alignment and uniformity, Zhang et al. (2023) propose matrix information theory, and design a matrix-theoretical self-supervised framework. Unfortunately, NCL demonstrates effectiveness only for CV and NLP tasks where the data is dense and the training signals are based on explicit feedback, while its effectiveness for SR is unexplored. Furthermore, the sparsity of the recommendation data leads to a higher risk of representation collapse for NCL.

## 3 PRELIMINARIES

**Data.** We focus on sequential recommendation (SR). In an SR dataset $\mathcal{D}$, a data sample is a sequence of interacted items $x$ (sorted by timestamps) from a user's history. We denote a list of interacted items as $x = [x_1, x_2, ..., x_l]$. For $x$, its ground-truth label is the item at the next time step: $y = x_{l+1}$. Each element in $x$ and $y$ belong to the item scope $\mathcal{I}$: $x_i \in \mathcal{I}$.

**Model.** Classic sequential recommenders are ID-based models that represent items with discrete item IDs. They take sequences of item IDs as inputs, and predict new item IDs as recommendations.

Given $x$, an ID-based recommender computes scores for all items from $\mathcal{I}$, and selects items with highest scores for recommendations. In comparison, text-based recommenders represent items with textual descriptions (e.g., title, category, reviews), and generate recommended items with free-form texts (e.g., item titles). In this work, we use a text-based model to build NCL-SR for the generality of textual features. For clarity and consistency, we also use $x = [x_1, x_2, ..., x_l]$ to denote the input texts of a text-based recommender, where each element $x_i$ is the textual description of an item.

**Contrastive Learning.** Existing CL-based recommendation methods usually employ the following contrastive loss to train recommendation models (Xie et al., 2022; Zhang et al., 2024; Shi et al., 2024). Formally, for a batch of training data $X$, the contrastive loss is computed with:

$$\mathcal{L}_{\text{CL}} = \frac{1}{|X|} \sum_{x^{(a)} \sim X} -\log \frac{s(x^{(a)}, x^{(p)})}{s(x^{(a)}, x^{(p)}) + \sum_{k=1}^{N} s(x^{(a)}, x_k^{(n)})}, \tag{1}$$

where $x^{(a)}$ denotes an anchor training sample within the batch, $x^{(p)}$ denotes a positive sample of $x^{(a)}$, and $x^{(n)}$ denotes a negative sample of $x^{(a)}$ (assume $N$ negative samples for $x^{(a)}$). $s$ represents an arbitrary similarity scoring function (e.g., cosine similarity) in a canonical form. However, CL suffers from high computational costs due to its reliance on negative samples and is sensitive to data augmentation. The above limitations of CL motivate our NCL framework.

## 4 METHODOLOGY

In this work, we propose a novel NCL-SR framework via preference-preserving profile generation to promote uniformity and alignment in user representations.

*Uniformity* ensures that that individual user preferences are preserved as much as possible in the user representations. However, within an NCL framework, the absence of negative samples makes it challenging to learn uniformity from sparse recommendation data. This is because such sparsity increases the risk of representation collapse (Guo et al., 2023) , where different user preferences are mapped nearby in the latent space. Consequently, it becomes difficult for the recommender to capture unique preferences of different users and generate meaningful recommendations. *Alignment* enforces that two users with similar preferences should be mapped to nearby representations, which makes representations robust against undesired noise factors. Nevertheless, the inconsistency within the training data hinders learning such alignment (e.g., implicit feedback, inconsistent user preferences in the augmented data). These inconsistencies confuse the model, when the model is trained to align profiles that reflect different, conflicting preferences for the same user.

Therefore, we design the preference-preserving profile generation to generate high-quality positive user profiles to facilitate non-contrastive training. To address representation collapse, our approach generates diversified positive samples. This helps the model explore distinct portions of the representation space, reducing the risk of representation collapse. In addition, with the reference-preserved positive samples, the data inconsistency issue is effectively addressed.

### 4.1 PREFERENCE-PRESERVING PROFILE AUGMENTATION VIA DIFFERENTIAL PRIVACY

To build the NCL framework for SR, we first propose the preference-preserving user profile generation. Inspired by Differential Privacy (DP) (Dwork et al., 2006), our approach creates augmented user profiles that retain original user preferences with provable guarantees. The preference-preserved augmentation generates diverse and consistent positive samples to overcome the representation collapse and inconsistency within the training data, contributing to better alignment and uniformity of the learned user representations. For a better understanding of our preference-preserving profile generation, we begin by formally defining preference-preserving augmentation. We provide further insights about Differential Privacy (DP) in Appendix A.

**Definition 1 (Preference-Preserving Augmentation)** *For a user profile $x$, its augmented view $x^{'}$ is considered preference-preserving if $x^{'}$ leads to the same expected purchase, or the expected top-1-ranked item remains the same, when $x'$ is fed into the same recommendation mechanism $\mathcal{M}$.*

With the definition of preference-preserving augmentation, we then present how DP guarantees preference-preservation in Theorem 1.

**Theorem 1 (Guaranteed Preference-Preservation with Differential Privacy)** *Suppose a recommendation mechanism $\mathcal{M}$ satisfies differential privacy with the privacy parameter $\epsilon$ (i.e., $\epsilon$-DP). For any user sequence $x$, if $\mathcal{M}$ successfully recommends the ground-truth item $y$, which is the $j$-th item within the item scope $\mathcal{I}$: $y := x_j$, and the predictive score for the ground-truth item is greater than the second-largest runner-up score of another item with a small multiplicative factor $e^{2\epsilon}$:*

$$\mathbb{E}(\mathcal{M}(x)_j) > e^{2\epsilon} \max_{k:k \neq j} \mathbb{E}(\mathcal{M}(x)_k), \tag{2}$$

*then for its augmented view $x'$, its predicted top-1 item remains the same if there is only limited modification from $x$ to $x'$:*

$$\mathbb{E}(\mathcal{M}(x')_j) > \max_{k:k \neq j} \mathbb{E}(\mathcal{M}(x')_k). \tag{3}$$

Note that Theorem 1 is a specific adaptation of a general DP property for our SR problem. The proof is adapted from (Wang et al., 2021), which we relegate to Appendix A. Since the recommendation data considered in this work is discrete (i.e., item texts), we use the DP exponential mechanism to build our DP augmentation pipeline, which is formally defined below.

**Definition 2 (Exponential Mechanism)** *The exponential mechanism $\mathcal{M}_E(u, \Delta_u, \hat{P})$ is characterized by three components: a scoring function $u$, the sensitivity $\Delta_u$ and a sampling distribution $\hat{P}$. The scoring function $u(X, X_c)$ computes scores for each pair of inputs $(X)$ and a candidate $(X_c)$. The sensitivity $\Delta_u$ is defined as $\Delta_u := \max_{X_c} \max_X |u(X, X_c) - u(X', X_c)|$. The exponential mechanism $\mathcal{M}_E(u, \Delta_u, \hat{P})$ is $\epsilon-$differentially private if it outputs the candidate $X_c$ with probability $\hat{P}$ proportional to $e^{\frac{\epsilon u(X, X_c)}{2\Delta_u}}$.*

**User Profile Augmentation via Exponential Mechanism.** When implementing the exponential mechanism for NCL-SR, we first craft a set of synonym dictionary for all items. The synonym items will be used to perturb the original user history following the exponential mechanism. For a specific item text $x_i \in \mathcal{I}$, its synonym items are other items with top-$k$ highest cosine similarity in the representation space. Formally, the synonym set $\mathcal{S}(x_i)$ of an item $x_i$ with top-$k$ similarity is computed with:

$$\mathcal{N}_k(x_i) = \arg \max_{\mathcal{N} \subset \mathcal{I}, |\mathcal{N}|=k, x_i \notin \mathcal{N}} \sum_{x_j \in \mathcal{N}} \text{CosSim}(f(x_i), f(x_j)), \tag{4}$$

where $f$ represents a text-based recommender model that encodes user/item texts into latent representations. After replacing randomly selected items within $x = [x_1, x_2, ..., x_l]$ with their synonyms, we obtain a set of candidate user profiles $X' = \{x'\}$. In terms of the exponential mechanism, the scoring function $u$ is then defined as the cosine similarity between embedded user profile $x$ and any other candidate user profiles $x'$:

$$u(x, x') = e^{\text{CosSim}(f(x), f(x'))} \tag{5}$$

Note that the candidate user profiles $x'$ are perturbed compared to the original user profile $x$, and the perturbations can be multiple on different positions. With the definition of Equation 5, the sensitivity of the utility score $u(x, x')$ is $\Delta_u = e - 1/e$, with $e$ being maximum of $u(x, x')$ and $1/e$ being the minimum. Finally, the exponential mechanism selects and outputs $x'$ with sampling probability $e^{\frac{\epsilon u(t, t')}{2\Delta_u}}$. For computational efficiency, the exponential mechanism returns the expected **latent representation** of $X'$ by computing the weighted sum of $x'$, with the importance weight for each $x'$ being the normalized sampling probability:

$$\hat{z_{X'}} = \sum_{x' \in X'} \hat{P_{x'}} \cdot f(x'), \quad \text{where} \quad \hat{P_{x'}} = \frac{e^{\frac{\epsilon u(x, x')}{2\Delta_u}}}{\sum_{x'' \in X'} e^{\frac{\epsilon u(x, x'')}{2\Delta_u}}}. \tag{6}$$

To this end, we obtain the augmented user profile for a training user. The DP user profiles effectively preserve the original user preferences and serve as the foundation for the NCL framework.

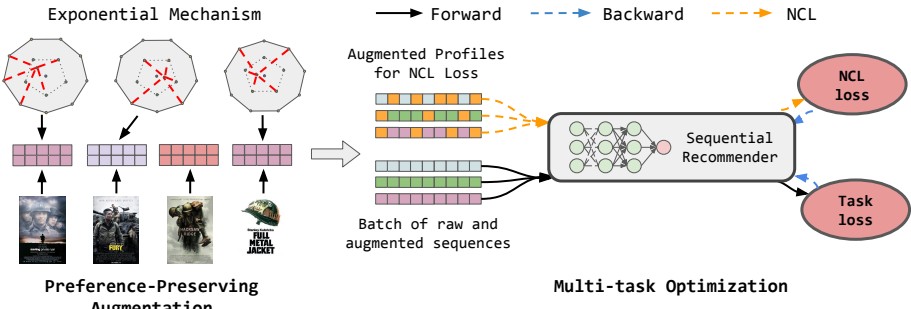

Figure 1: The overview of the proposed NCL-SR. NCL-SR first performs preference-preserving augmentation to generate high-quality positives. Then, NCL-SR optimizes a combined training loss of NCL losses and the main task loss (e.g., cross-entropy or other ranking loss).

Note that a critical caveat here is that the search space grows exponentially w.r.t. the length of user histories if the augmentation is defined at user-level: $\mathcal{O}(k^l)$. This is because each item $x_i \in x = [x_1, x_2, ..., x_l]$ has $k$ replacements. As such, there are $k^l$ possible candidate user profiles for the exponential mechanism to sample from, i.e., $|X'| = k^l$. Given the sparsity of recommendation data, computing DP augmentations for user sequences introduces prohibitive computation complexity and noises. Therefore, in our experiments, we re-define Equation 5 and Equation 6 at item-level, successfully reducing the complexity to be linear: $\mathcal{O}(k \cdot l)$. To see this, we highlight that when applying the exponential mechanism to each item $x_i \in x = [x_1, x_2, ..., x_l]$, the exponential mechanism just needs to sample from $k$ candidates, and repeat for all items within the history $l$ times, leading to $\mathcal{O}(k \cdot l)$. More details (guarantees, proofs, and implementation) about the efficient design are documented in Appendix B.

An illustrative example of DP profile generation is shown in Figure 1. In Figure 1, each polygon represents the synonym set for an item. The vertices within a polygon represent the embeddings of the synonym items w.r.t. the original item. For instance, for the first item (i.e., the movie "Save the Private Ryan ") of the user's history, our method first encodes this item into an embedding vector. Then, we compute the synonym set for this item with Equation 4. Therefore, all vertices in the polygon represent the embedded items. Note that there is no corresponding polygon for the third item in Figure 1, because our approach only perturbs a part of the user profiles, which reduces computation costs of the exponential mechanism.

Note this efficient design is feasible and remains DP, because of the two key properties of DP. First, DP has the post-processing property: any computation applied to the output of a DP algorithm remains DP. As such, if the augmentation operation is DP, then the recommendation process based on the augmented user profile is also DP. Second, the expected output stability property of DP reduces the computational costs of DP augmentation. This property allows us to generate an expected augmented profile for each user to preserve user preferences in the augmented profiles, instead of generating massive augmented samples for Monte-Carlo sampling.

## 4.2 OVERVIEW OF NCL-SR

With the proposed preference-preserving profile generation, we then introduce the overall NCL-SR. As shown in Figure 1, our proposed NCL-SR consists of 3 components.

The first component is a text-based recommender $f$ that encodes user/item texts and returns relevance scores over candidate items. For any user sequence $x$, the recommender $f$ encodes user/item texts into representations and then returns a probability distribution over the item scope $\mathcal{I}$:

$$P = P(f, x, \mathcal{I}) = \text{Softmax}\left(\left[\text{CosSim}(q, p_1), \text{CosSim}(q, p_2), ..., \text{CosSim}(q, p_{|\mathcal{I}|})\right]\right), \quad (7)$$

where $q = f_T(x)$ is the embedded user profile, and $p_j = f_T(x_j), x_j \in \mathcal{I}$ is the embedded description of an item $x_j$. The items associated with highest probabilities are retrieved as recommendations.

The second component is multi-task optimization, where we optimize the model w.r.t. the ranking task, alignment learning task, and uniformity learning task. The multi-task optimization benefits recommendation performance, because both the recommendation task and non-contrastive training are modeling user/item relationships in the representation space. To design a general NCL framework for sequential recommendation, we adopt Matrix Cross Entropy (MCE) (Zhang et al., 2023), which generalizes multiple non-contrastive losses including SimSiam (Chen & He, 2021), Barlow Twin (Zbontar et al., 2021), MEC (Liu et al., 2022), etc. Formally, the matrix cross entropy between two matrices $U$ and $V$ is defined as

$$\text{MCE}(U, V) = \text{tr}(-U \log V + V), \tag{8}$$

where $\log$ denotes the principal matrix logarithm (Higham, 2008) and is approximated via Taylor expansion. Based on the definition of matrix cross entropy, one may promote uniformity in representation learning by aligning the covariance matrix of representations with the identity matrix:

$$\mathcal{L}_{\text{uniform}}(Z, Z^{'}) = \text{MCE}\big(\frac{1}{d}I_d, C(Z, Z^{'})\big), \tag{9}$$

where $Z = f(X)$ and $Z^{'} = f(X^{'})$ denote encoded, $d$-dimensional, $l2$-normalized representations of batched user profiles and their preference-preserving augmentation, respectively. $C(Z, Z^{'})$ represents the centered covariance matrix of $Z$ and $Z^{'}$: $C(Z, Z^{'}) = \frac{1}{B}ZH_BZ^{'T}, H_B = I_B - \frac{1}{B}1_B1_B^T$. Intuitively, minimizing $\mathcal{L}_{\text{uniform}}(Z, Z^{'})$ promotes the uniformity in representation learning, because the minimization makes the covariance matrix of the embedded (augmented) user representations to be close to the identity matrix, which increases the entropy of the encoded representations.

Note that Equation 9 as well as other existing non-contrastive methods only promotes uniformity in representation learning. However, it has been widely shown that learning aligned user representations also plays a vital role in improving model generalization for sequential recommendation. This is because aligned representations are more robust against the noises caused by data inconsistency (Wang et al., 2022b; Xie et al., 2022; Qiu et al., 2022). Therefore, we propose to integrate another alignment loss to train the model w.r.t. the alignment learning loss:

$$\mathcal{L}_{\text{align}}(Z, Z^{'}) = -\text{tr}(C(Z, Z^{'})) + \gamma \cdot \text{MCE}\big(C(Z, Z), C(Z^{'}, Z^{'})\big). \tag{10}$$

Intuitively, since $Z$ and $Z^{'}$ are derived from the same group of users as in Equation 9, minimizing $\mathcal{L}_{\text{align}}(Z, Z^{'})$ enforces the similarity of the learned representations from the same user. To this end, the multi-task optimization is a combination of losses of three tasks:

$$\mathcal{L}_{\text{NCL-SR}} = \mathcal{L}_{\text{task}} + \lambda_1 \cdot \mathcal{L}_{\text{uniform}} + \lambda_2 \cdot \mathcal{L}_{\text{align}}. \tag{11}$$

The third component of NCL-SR is the preference-preserving profile generation introduced in the previous section. We highlight that both Equation 9 and Equation 10 require the augmented views of original user profiles. In other words, generating high-quality augmented data is the prerequisite for non-contrastive training. This component can effectively preserve the user preferences in the diverse, augmented user profiles, which overcomes the representation collapse issue and the inconsistency within the training data.

## 5 EXPERIMENTS

### 5.1 EXPERIMENTAL SETUP

**Datasets.** We adopt 6 public benchmark recommendation datasets to evaluate NCL-SR: Beauty, Games, Sports, Toys, Office and Auto (He & McAuley, 2016; McAuley et al., 2015)[*]. These datasets cover different application domains, and are characterized with different sparsity and sequence lengths. Following Yue et al. (2022), we use the 5-core processing to filter out infrequent items and users. To simulate the data sparsity and cold-start users, the split ratio of the training, validation, and test sets is 2:2:6. Such a split ratio is selected because we aim to explore the model's performance under a limited quantity of training data and the model's generalization on cold-start users, following the recent studies Wu et al. (2024); Qian et al. (2020); Wang et al. (2022a); Lin et al. (2025). More details about dataset statistics are in summarized Appendix C[†].

---

[*]The datasets are available at http://snap.stanford.edu/data/amazon/productGraph/categoryFiles/.

[†]Our code is available at https://github.com/huiminzeng/NCL-SR.git

| Dataset | Metric | ID-Based | | | | Text-Based | | | | Improv. |
| | | BERT4Rec | SASRec | NARM | LRURec | P5 | UniSRec | RecFmr. | NCL-SR | (%) |
|---|---|---|---|---|---|---|---|---|---|---|
| **Beauty** | R@10 ↑ | 0.0429 | 0.0599 | 0.0493 | 0.0715 | 0.0227 | 0.0437 | 0.0729 | **0.0791** | 8.50% |
| | N@10 ↑ | 0.0232 | 0.0362 | 0.0281 | 0.0439 | 0.0133 | 0.0249 | 0.0407 | **0.0440** | 0.23% |
| | R@20 ↑ | 0.0617 | 0.0812 | 0.0719 | 0.0931 | 0.0282 | 0.0656 | 0.1035 | **0.1135** | 9.66% |
| | N@20 ↑ | 0.0279 | 0.0415 | 0.0338 | 0.0493 | 0.0147 | 0.0304 | 0.0484 | **0.0526** | 6.69% |
| **Games** | R@10 ↑ | 0.0564 | 0.0934 | 0.0661 | 0.1087 | 0.0120 | 0.0691 | 0.0867 | **0.1140** | 4.88% |
| | N@10 ↑ | 0.0294 | 0.0490 | 0.0350 | 0.0606 | 0.0063 | 0.0365 | 0.0465 | **0.0611** | 0.83% |
| | R@20 ↑ | 0.0932 | 0.1323 | 0.1036 | 0.1540 | 0.0196 | 0.1024 | 0.1339 | **0.1683** | 9.29% |
| | N@20 ↑ | 0.0386 | 0.0587 | 0.0444 | 0.0720 | 0.0083 | 0.0449 | 0.0583 | **0.0748** | 3.89% |
| **Sports** | R@10 ↑ | 0.0195 | 0.0259 | 0.0272 | 0.0327 | 0.0057 | 0.0203 | 0.0331 | **0.0441** | 33.2% |
| | N@10 ↑ | 0.0111 | 0.0154 | 0.0147 | 0.0201 | 0.0028 | 0.0115 | 0.0172 | **0.0237** | 17.9% |
| | R@20 ↑ | 0.0291 | 0.0330 | 0.0389 | 0.0464 | 0.0085 | 0.0317 | 0.0518 | **0.0660** | 27.4% |
| | N@20 ↑ | 0.0135 | 0.0172 | 0.0177 | 0.0235 | 0.0036 | 0.0143 | 0.0218 | **0.0292** | 24.3% |
| **Toys** | R@10 ↑ | 0.0278 | 0.0379 | 0.0299 | 0.0655 | 0.0253 | 0.0382 | 0.0845 | **0.0941** | 11.4% |
| | N@10 ↑ | 0.0166 | 0.0226 | 0.0173 | 0.0420 | 0.0139 | 0.0221 | 0.0475 | **0.0537** | 13.1% |
| | R@20 ↑ | 0.0379 | 0.0507 | 0.0444 | 0.0858 | 0.0319 | 0.0548 | 0.1164 | **0.1286** | 10.5% |
| | N@20 ↑ | 0.0192 | 0.0258 | 0.0210 | 0.0471 | 0.0156 | 0.0262 | 0.0555 | **0.0623** | 12.3% |
| **Office** | R@10 ↑ | 0.0453 | 0.0524 | 0.0654 | 0.0926 | 0.0528 | 0.0629 | 0.0645 | **0.1047** | 13.1% |
| | N@10 ↑ | 0.0231 | 0.0325 | 0.0333 | 0.0496 | 0.0255 | 0.0300 | 0.0350 | **0.0534** | 7.66% |
| | R@20 ↑ | 0.0782 | 0.0996 | 0.1155 | 0.1462 | 0.0915 | 0.1018 | 0.1025 | **0.1625** | 11.2% |
| | N@20 ↑ | 0.0314 | 0.0445 | 0.0459 | 0.0631 | 0.0351 | 0.0398 | 0.0445 | **0.0681** | 7.92% |
| **Auto** | R@10 ↑ | 0.0415 | 0.0439 | 0.0585 | 0.1220 | 0.0431 | 0.0646 | 0.1085 | **0.1354** | 11.0% |
| | N@10 ↑ | 0.0202 | 0.0216 | 0.0287 | 0.0632 | 0.0216 | 0.0355 | 0.0571 | **0.0714** | 13.0% |
| | R@20 ↑ | 0.0732 | 0.0780 | 0.1024 | 0.1829 | 0.0866 | 0.1098 | 0.1768 | **0.2085** | 14.0% |
| | N@20 ↑ | 0.0282 | 0.0304 | 0.0396 | 0.0785 | 0.0325 | 0.0469 | 0.0745 | **0.0899** | 14.5% |

Table 1: Main results on recommendation performance of different SR models. The best results are highlighted in bold and the second best results are highlighted with underline.

**Baselines and Implementation.** There are two sets of baselines in our experiments. The first group of baselines are state-of-the-art SR models, including ID-based models (i.e., NARM (Li et al., 2017), LRURec (Yue et al., 2023), BERT4Rec (Sun et al., 2019), SASRec (Kang & McAuley, 2018)) and text-based SR model: P5 (Geng et al., 2022), RecFmr (Li et al., 2023) and UniSR$_T$ (Hou et al., 2022). The second group baselines are state-of-the-art CL-based recommendation methods, including CLS4Rec (Xie et al., 2022), CoSeRec (Liu et al., 2021), EC4Rec (Liu et al., 2021), DUORec (Qiu et al., 2022) and SCL (Shi et al., 2024). Note that some of the CL-based baselines are developed based on item IDs, and we modify such baselines into the text-based setting for a fair comparison. We use E5 (`e5-base-v2`) (Wang et al., 2022c) to implement NCL-SR and CL-based baselines. When generating DP augmentations, we randomly perturb 3 items for each training user from datasets. For evaluation, we use normalized discounted cumulative gain (NDCG@N) and recall (Recall@N) with $N \in [10, 20]$. The predictions are ranked against all items in the dataset.

## 5.2 COMPARING AGAINST SR MODELS

The first set of experiments is conducted to compare NCL-SR against state-of-the-art sequential recommenders. The performance results are reported in Table 1. The last column of Table 1 is relative improvement NCL-SR compared to the best-performing baseline method (i.e., Improv.). We observe: (1) NCL-SR consistently outperforms baseline methods across all metrics and datasets, with an average performance improvement of 11.93% compared to the second best method. Such an observation demonstrates the generality of NCL-SR, regardless of data domains. (2) The performance gains of NCL-SR are more pronounced on the sparsest dataset (i.e., Sports), where NCL-SR can go up to 33.2% on Recall@10. This demonstrates the substantial benefits of applying NCL-SR in sparse data domains. (3) NCL-SR generally demonstrates better retrieval performance (i.e., Recall) than ranking performance (i.e., NDCG). For instance, there is a significant increase of 16.7% in the average Recall@10 scores with NCL-SR, while the relative improvement on NDCG@10 is slightly lower (8.78%). This is expected, as our text-based model has a strong ability to understand and match the semantic meanings of item descriptions, which provides advantages for the retrieval task. We also implement the SR models with E5 for additional comparison, whose results are reported in Appendix F.

| Dataset | Metric | CL-Based | | | | | NCL | Improv. |
| | | CLS4Rec | CoSeRec | DUORec | EC4Rec | SCL | NCL-SR | (%) |
|---|---|---|---|---|---|---|---|---|
| **Beauty** | R@10 ↑ | 0.0690 | 0.0698 | 0.0706 | 0.0628 | 0.0709 | **0.0791** | 11.6% |
| | N@10 ↑ | 0.0367 | 0.0375 | 0.0383 | 0.0344 | 0.0375 | **0.0440** | 14.9% |
| | R@20 ↑ | 0.1008 | 0.1007 | 0.1052 | 0.0971 | 0.1026 | **0.1135** | 7.89% |
| | N@20 ↑ | 0.0447 | 0.0453 | 0.0470 | 0.0430 | 0.0456 | **0.0526** | 11.9% |
| **Games** | R@10 ↑ | 0.0992 | 0.1057 | 0.1041 | 0.0998 | 0.1012 | **0.1140** | 7.85% |
| | N@10 ↑ | 0.0521 | 0.0553 | 0.0543 | 0.0524 | 0.0533 | **0.0611** | 10.5% |
| | R@20 ↑ | 0.1458 | 0.1560 | 0.1545 | 0.1516 | 0.1499 | **0.1683** | 7.89% |
| | N@20 ↑ | 0.0638 | 0.0680 | 0.0669 | 0.0654 | 0.0655 | **0.0748** | 10.0% |
| **Sports** | R@10 ↑ | 0.0325 | 0.0327 | 0.0332 | 0.0337 | 0.0314 | **0.0441** | 30.9% |
| | N@10 ↑ | 0.0173 | 0.0169 | 0.0174 | 0.0180 | 0.0163 | **0.0237** | 31.7% |
| | R@20 ↑ | 0.0529 | 0.0501 | 0.0504 | 0.0511 | 0.0473 | **0.0660** | 24.8% |
| | N@20 ↑ | 0.0223 | 0.0213 | 0.0217 | 0.0223 | 0.0203 | **0.0292** | 30.9% |
| **Toys** | R@10 ↑ | 0.0898 | 0.0868 | 0.0882 | 0.0871 | 0.0895 | **0.0941** | 4.79% |
| | N@10 ↑ | 0.0511 | 0.0490 | 0.0501 | 0.0497 | 0.0511 | **0.0537** | 5.09% |
| | R@20 ↑ | 0.1221 | 0.1199 | 0.1232 | 0.1201 | 0.1249 | **0.1286** | 2.96% |
| | N@20 ↑ | 0.0592 | 0.0573 | 0.0589 | 0.0580 | 0.0600 | **0.0623** | 3.83% |
| **Office** | R@10 ↑ | 0.0974 | 0.0967 | 0.0974 | 0.0913 | 0.0980 | **0.1047** | 6.84% |
| | N@10 ↑ | 0.0498 | 0.0487 | 0.0518 | 0.0465 | 0.0502 | **0.0534** | 3.09% |
| | R@20 ↑ | 0.1471 | 0.1561 | 0.1497 | 0.1459 | 0.1500 | **0.1625** | 4.10% |
| | N@20 ↑ | 0.0623 | 0.0635 | 0.0647 | 0.0603 | 0.0634 | **0.0681** | 5.26% |
| **Auto** | R@10 ↑ | 0.1159 | 0.1171 | 0.1098 | 0.1073 | 0.1122 | **0.1354** | 15.6% |
| | N@10 ↑ | 0.0575 | 0.0576 | 0.0572 | 0.0535 | 0.0559 | **0.0714** | 24.0% |
| | R@20 ↑ | 0.1829 | 0.1902 | 0.1927 | 0.1793 | 0.1780 | **0.2085** | 8.20% |
| | N@20 ↑ | 0.0742 | 0.0761 | 0.0781 | 0.0713 | 0.0724 | **0.0899** | 15.1% |

Table 2: Main results on recommendation performance of different CL and NCL methods.

## 5.3 COMPARING AGAINST CL-BASED METHODS

We compare NCL-SR against state-of-the-art CL-based SR methods. The performance results are reported in Table 2. We observe: (1) our method demonstrates consistent performance improvements over the baseline CL methods across all metrics and datasets, with an average improvement of 12.48% compared to the second best method. (2) On the most sparse dataset (i.e., Sports), NCL-SR still achieves the most performance gain, suggesting that NCL is more efficient that CL-based methods in terms of learning generalizable representations from sparse data. Recall Due to the space limit, we provide additional experiments on exploring the relationship between CL, NCL, and data augmentation in Appendix D, and compare their memory consumption in Appendix G.

## 5.4 ABLATION STUDY

**Alignment and Uniformity.** We conduct an ablation to understand the effects of learning aligned and uniform representations for our SR task. In particular, we set $\lambda_1 = 0$ in Equation 11 to remove the uniformity (Ours w/o Uni.) or set $\lambda_2 = 0$ to remove the alignment (Ours w/o Align.), respectively. The experimental results are reported in Table 3. From Table 3, it is observed that in general, both alignment and uniformity play a vital role in learning generalizable user representations for SR. Specifically, when removing the alignment loss in Equation 11, the averaged performance on Recall@10 and NDCG@10 across all datasets would drop by 6.10% and 6.47%, respectively. In comparison, there is an average drop of 4.00% on Recall@10 and 3.92% on NDCG@10 if uniformity is removed. Furthermore, better recommendation performance can be observed when only learning the aligned user representations. For instance, on Beauty, there is a noticeable increase in both Recall@20 and NDCG@20 without uniformity loss. To this end, we conclude that promoting alignment may be more important than uniformity in learning generalizable representations for SR applications. This further necessitates the design and merit of the proposed preference-preserving augmentation, which addresses the preference inconsistency caused by ad-hoc augmentations.

**DP augmentation and NCL loss.** We then conduct another ablation to investigate the effects of the proposed DP augmentation and the NCL losses. Specifically, we mask out either module by (1) replacing our DP augmentation with other data augmentation operations that are randomly sampled from the CL-based baselines (Ours w/o DP Aug.) or (2) replacing the NCL loss with the

| Variants | Metric | Beauty | Games | Sports | Toys | Office | Auto |
|---|---|---|---|---|---|---|---|
| | | Recall/NDCG | Recall/NDCG | Recall/NDCG | Recall/NDCG | Recall/NDCG | Recall/NDCG |
| **NCL-SR** | @10 | **0.0791 / 0.0440** | **0.1140 / 0.0611** | **0.0441 / 0.0237** | **0.0941 / 0.0537** | 0.1047 / **0.0534** | **0.1354 / 0.0714** |
| **(Ours)** | @20 | 0.1135 / 0.0526 | **0.1683 / 0.0748** | **0.0660 / 0.0292** | 0.1286 / **0.0623** | **0.1625 / 0.0681** | 0.2085 / **0.0899** |
| (1) Ours | @10 | 0.0761 / 0.0408 | 0.1133 / 0.0590 | 0.0384 / 0.0205 | 0.0916 / 0.0504 | 0.0954 / 0.0524 | 0.1293 / 0.0691 |
| w/o Align. | @20 | 0.1124 / 0.0499 | 0.1673 / 0.0725 | 0.0606 / 0.0261 | 0.1270 / 0.0594 | 0.1580 / 0.0680 | 0.1988 / 0.0865 |
| (2) Ours | @10 | 0.0784 / 0.0437 | 0.1128 / 0.0592 | 0.0404 / 0.0221 | 0.0936 / 0.0529 | 0.0983 / 0.0519 | 0.1280 / 0.0661 |
| w/o Uni. | @20 | **0.1149** / 0.0529 | 0.1667 / 0.0728 | 0.0630 / 0.0277 | **0.1304** / 0.0622 | 0.1519 / 0.0652 | **0.2171** / 0.0885 |
| (1) Ours | @10 | 0.0717 / 0.0394 | 0.1029 / 0.0552 | 0.0364 / 0.0189 | 0.0911 / 0.0517 | **0.1053** / 0.0530 | 0.1329 / 0.0705 |
| w/o DP Aug. | @20 | 0.1046 / 0.0477 | 0.1520 / 0.0675 | 0.0543 / 0.0234 | 0.1264 / 0.0606 | 0.1554 / 0.0656 | 0.2049 / 0.0886 |
| (2) Ours | @10 | 0.0727 / 0.0394 | 0.1014 / 0.0529 | 0.0369 / 0.0198 | 0.0834 / 0.0456 | 0.0980 / 0.0527 | 0.1268 / 0.0678 |
| w/o NCL | @20 | 0.1077 / 0.0481 | 0.1529 / 0.0659 | 0.0553 / 0.0245 | 0.1160 / 0.0538 | 0.1465 / 0.0647 | 0.2073 / 0.0879 |

Table 3: Ablation Study on different components of NCL-SR.

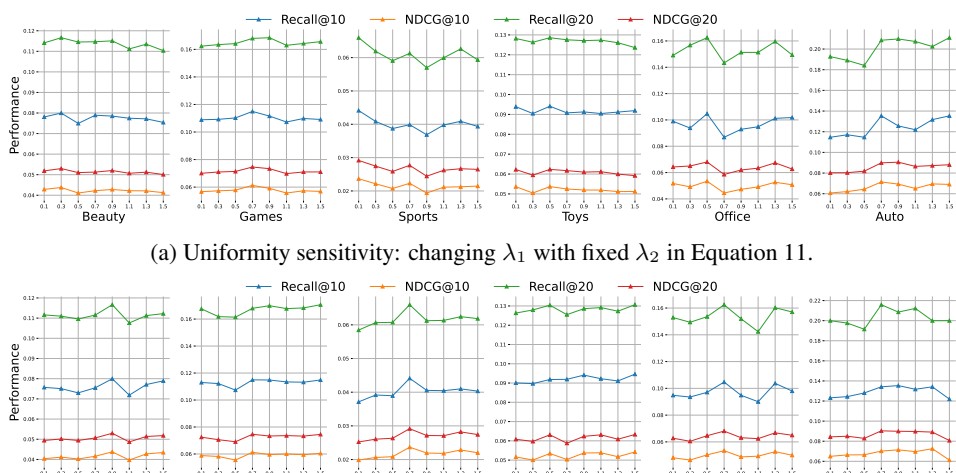

(a) Uniformity sensitivity: changing $\lambda_1$ with fixed $\lambda_2$ in Equation 11.

(b) Alignment sensitivity: changing $\lambda_2$ with fixed $\lambda_1$ in Equation 11.

Figure 2: Sensitivity Analysis w.r.t. Alignment and Uniformity.

contrastive loss (Ours w/o NCL). The results are reported in Table 3. It is observed that removing either module generally leads to performance degradation, indicating that both DP augmentation and NCL representation learning are necessary for improving recommendation performance.

### 5.5 SENSITIVITY ANALYSIS

We finally study the sensitivity of NCL-SR w.r.t. the key hyperparameters $\lambda_1$ and $\lambda_2$ in Equation 11, where $\lambda_1$ controls the strength of uniformity and $\lambda_2$ controls the strength of alignment. When performing sensitivity analysis, we select and fix the best $\lambda_1$, and change $\lambda_2$ to understand the sensitivity w.r.t. alignment. Similarly, $\lambda_2$ is fixed and $\lambda_1$ is changed for the sensitivity analysis w.r.t. uniformity. The performance is visualized in Figure 2. We observe that there exist different optimal configurations for different datasets. Additional sensitivity analysis can be found at Appendix E.

## 6 CONCLUSION

In this work, we design the very first Non-Contrastive Learning framework for Sequential Recommendation: NCL-SR. In particular, we present a principled data augmentation operation to produce preference-preserving user profiles, based on which we define uniformity and alignment losses to learn user representations. Our investigations reveal that learning uniform and aligned user representations play a vital role in the SR task, and alignment may be more important than uniformity. Our extensive experimental results suggest NCL-SR consistently exhibits superior performance over state-of-the-art SR models and CL-based SR methods.

ACKNOWLEDGMENT

We thank Kiran Koshy Thekumprampil [*] and Mukund Seshadri [†] for their support in model training infrastructure. We also thank all the anonymous reviewers for their helpful comments.

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
