# OpenReview forum: "A Non-Contrastive Learning Framework for Sequential Recommendation with Preference-Preserving Profile Generation"
_ICLR.cc/2025/Conference — ICLR 2025 Poster_

### Official Review · Reviewer_A3iE · 2024-10-26

**Soundness:** 3
**Presentation:** 3
**Contribution:** 4
**Rating:** 8
**Confidence:** 4

**Summary:**

The paper introduces the NCL-SR, which is designed to reduce the computational burden found in traditional Contrastive Learning apporaches by eliminating the need for negative sampling. CL methods generally require negative samples to avoid representation collapse, but this comes with high computational costs. The main contribution of NCL-SR is its preference-preserving profile generation technique, which uses differential privacy to create diverse yet consistent positive user profile samples that retain the user’s preferences. This helps to overcome challenges such as representation collapse and preference inconsistency.

**Strengths:**

- This paper solves a important question in traditional SR field, that CL approaches cost quite a lot computational resources. By adopting the profile augmentation method proposed in this paper, the performance of SR can even higher than those traditional CL methods.

- The authors provides theoritial guarantees for the uniformity and alignment.

**Weaknesses:**

- Since the main motivation of NCL is to reduce the cost of traditional CL methods, I believe the efficiency study is needed.
- When comparing with other SR models (besides CL methods), I believe they should also be implemented with e5 for a fairer comparison.

**Questions:**

- The authors claim that the user profile generation is inspired by differential privacy, I wonder how it inspired the proposed framework in details? Maybe some addtional literature review might help to explain.

---

> ### Author Response · Authors · 2024-11-25
> **Response to Reviewer A3iE (Part 1)**
>
> We greatly appreciate Reviewer A3iE for the valuable feedback and suggestions. We are encouraged that the reviewer finds our paper important. To address the reviewer’s questions, we provide detailed clarifications and discussions below.
>
> **W1: Since the main motivation of NCL is to reduce the cost of traditional CL methods, I believe the efficiency study is needed.**
>
> We appreciate this valuable comment. To address the reviewer’s concern, we provide the following clarifications and additional experimental results as follows:
>
> - From a theoretical point of view, the time complexity of CL is $\mathcal{O}(n^2)$, if the training batch size is n and the loss is computed with in-batch negative samples. In comparison, the time complexity of NCL is $\mathcal{O}(n)$, because there are no negative samples at all. Therefore, we mainly cite some related work [1,2,3,4] to support the statement that NCL is more computationally efficient than CL.
>
> - Furthermore, we conducted additional experiments to evaluate the computational cost and memory consumption of computing CL and NCL losses. In this set of experiments, we ensure that all methods use exactly the same training configuration. Moreover, we vary the batch size from 1 to 5. The CUDA version we use is 12.1. The GPU used to evaluate memory consumption is NVIDIA A40 with 46 GB memory. We report the results in the table below. In the Table below, we report the relative memory consumption between NCL and CL $\frac{\text{memory consumption of NCL}}{\text{memory consumption of CL}}$:
>
> | Batch Size | 1    | 2    | 3    | 4    | 5                |
> |------------|------|------|------|------|------------------|
> | NCL/CL     | 0.72 | 0.57 | 0.51 | 0.51 | CL out of memory |
>
> - We will add the above results and analysis in our final version.
>
> [1] Jure Zbontar, Li Jing, Ishan Misra, Yann LeCun, and St´ephane Deny. Barlow twins: Self-supervised learning via redundancy reduction. In International conference on machine learning, pp. 12310– 12320. PMLR, 2021.
>
> [2] Jean-Bastien Grill, Florian Strub, Florent Altch´e, Corentin Tallec, Pierre Richemond, Elena Buchatskaya, Carl Doersch, Bernardo Avila Pires, Zhaohan Guo, Mohammad Gheshlaghi Azar, et al. Bootstrap your own latent-a new approach to self-supervised learning. Advances in neural information processing systems, 33:21271–21284, 2020.
>
> [3] Jaejin Cho, Jes´us Villalba, Laureano Moro-Velazquez, and Najim Dehak. Non-contrastive self- supervised learning for utterance-level information extraction from speech. IEEE Journal of Selected Topics in Signal Processing, 16(6):1284–1295, 2022.
>
> [4] Zhijian Zhuo, Yifei Wang, Jinwen Ma, and Yisen Wang. Towards a unified theoretical understanding of non-contrastive learning via rank differential mechanism. arXiv preprint arXiv:2303.02387, 2023.

---

> ### Author Response · Authors · 2024-11-25
> **Response to Reviewer A3iE (Part 2)**
>
> **W2: When comparing with other SR models (besides CL methods), I believe they should also be implemented with e5 for a fairer comparison.**
>
> We appreciate this valuable comment. To address the reviewer’s comment, we conduct additional experiments as suggested by the reviewer. In particular, we implemented SR models with E5, where E5 is used as the embedding model to embed the item texts. We compare our method against the E5-based SR models in the table below. It is observed that our method can outperform the E5-based SR models. We also note that it is non-trivial to apply E5 to these models, because such ID-based models originally take item IDs as inputs, whereas E5 is a text-based model.
>
> - Beauty
>
> |      | BERT4Rec  (E5) | SASRec (E5) | NARM  (E5) | LRURec  (E5) | Ours   |
> |------|----------------|-------------|------------|--------------|--------|
> | R@10 | 0.0420         | 0.0504      | 0.0348     | 0.0446       | 0.0791 |
> | N@10 | 0.0226         | 0.0249      | 0.0189     | 0.0210       | 0.0440 |
> | R@20 | 0.0672         | 0.0784      | 0.0486     | 0.0614       | 0.1135 |
> | N@20 | 0.0290         | 0.0319      | 0.0224     | 0.0292       | 0.0526 |
>
> - Games
>
> |      | BERT4Rec  (E5) | SASRec (E5) | NARM  (E5) | LRURec  (E5) | Ours   |
> |------|----------------|-------------|------------|--------------|--------|
> | R@10 | 0.0739         | 0.1061      | 0.0422     | 0.0641       | 0.1140 |
> | N@10 | 0.0385         | 0.0556      | 0.0236     | 0.0351       | 0.0611 |
> | R@20 | 0.1150         | 0.1568      | 0.0563     | 0.0946       | 0.1683 |
> | N@20 | 0.0488         | 0.0682      | 0.0272     | 0.0429       | 0.0748 |
>
> - Sports
>
> |      | BERT4Rec  (E5) | SASRec (E5) | NARM  (E5) | LRURec  (E5) | Ours   |
> |------|----------------|-------------|------------|--------------|--------|
> | R@10 | 0.0287         | 0.0253      | 0.0105     | 0.0164       | 0.0441 |
> | N@10 | 0.0159         | 0.0126      | 0.0064     | 0.0095       | 0.0237 |
> | R@20 | 0.0431         | 0.0372      | 0.0144     | 0.0228       | 0.0660 |
> | N@20 | 0.0195         | 0.0156      | 0.0074     | 0.0111       | 0.0292 |
>
> - Toys
>
> |      | BERT4Rec  (E5) | SASRec (E5) | NARM  (E5) | LRURec  (E5) | Ours   |
> |------|----------------|-------------|------------|--------------|--------|
> | R@10 | 0.0379         | 0.0533      | 0.0127     | 0.0198       | 0.0941 |
> | N@10 | 0.0197         | 0.0268      | 0.0088     | 0.0133       | 0.0537 |
> | R@20 | 0.0576         | 0.0739      | 0.0158     | 0.0256       | 0.1286 |
> | N@20 | 0.0247         | 0.0320      | 0.0096     | 0.0147       | 0.0623 |
>
> - Office
>
> |      | BERT4Rec  (E5) | SASRec (E5) | NARM  (E5) | LRURec  (E5) | Ours   |
> |------|----------------|-------------|------------|--------------|--------|
> | R@10 | 0.0342         | 0.0307      | 0.0425     | 0.0466       | 0.1047 |
> | N@10 | 0.0170         | 0.0156      | 0.0210     | 0.0210       | 0.0534 |
> | R@20 | 0.0549         | 0.0434      | 0.0836     | 0.0782       | 0.1625 |
> | N@20 | 0.0222         | 0.0187      | 0.0313     | 0.0289       | 0.0681 |
>
> - Auto
>
> |      | BERT4Rec  (E5) | SASRec (E5) | NARM  (E5) | LRURec  (E5) | Ours   |
> |------|----------------|-------------|------------|--------------|--------|
> | R@10 | 0.0378         | 0.0439      | 0.0402     | 0.0512       | 0.1354 |
> | N@10 | 0.0206         | 0.0279      | 0.0207     | 0.0275       | 0.0714 |
> | R@20 | 0.0646         | 0.0537      | 0.0683     | 0.0817       | 0.2085 |
> | N@20 | 0.0282         | 0.0303      | 0.0227     | 0.0352       | 0.0899 |

---

> ### Author Response · Authors · 2024-11-25
> **Response to Reviewer A3iE (Part 3)**
>
> **Q1: The authors claim that the user profile generation is inspired by differential privacy, I wonder how it inspired the proposed framework in detail? Maybe some additional literature review might help to explain.**
>
> To address the reviewer's question, we clarify that the key intuition of using differential privacy (DP) for our preference-preserving profile generation is as follows:
> - From a high-level point of view, we regard each user profile in our recommendation problem as a database in the definition of DP. Consequently, each interacted item within a profile can be regarded as a record in a database. According to the definition of DP, if a mechanism satisfies DP, an observer analyzing its output cannot tell whether a particular data record was used in the computation. As such, applying DP to user profile generation, the recommender functions as the observer. By ensuring DP, the recommender cannot tell whether a particular item from the user's history was used to compute the item scores. Therefore, when we formulate the above process into our user profile generation, it is straightforward that: with limited perturbations, the generated user profile will not cause a significant change in the resulting scores if the entire generation satisfies DP.
> - In addition, the above process can also be prescribed by the mathematical definition of DP: $P[\mathcal{M}(X) \in Y ] \leq e^{\epsilon}P[\mathcal{M}(X^{'})\in Y]$. According the definition of DP, the output distribution of a DP mechanism will exhibit a limited change, if the input is perturbed with a limited budget.
>
> - Our approach is closely related to the findings in the cited [1,2].  In [1,2], DP  is used to provide certified robustness of classifiers against adversarial examples. That is, in these studies, the goal is to ensure that a classifier’s predictions remain consistent even if the input is deliberately perturbed. Inspired by [1,2], our method leverages DP to generate augmented user profiles that preserve user preferences. In the context of recommendation, this means that the model should produce consistent recommendations for the original user profile and the augmented one, if they represent the same user preference. Additionally, recent studies [3,4] have also explored the trade-off between the analysis accuracy and perturbations under DP. Such studies do not necessarily only focus on the privacy benefits of DP, but also aim to maintain high accuracy under noises or perturbations. In summary, inspired by [1,2,3,4], we propose to use DP to augment user profiles for recommendation systems. Our goal is to preserve user preferences during the augmentation process: a perturbed user profile should yield the same recommendations as its original counterpart. This consistency indicates that user preferences are well-preserved in the augmented data, ensuring high-quality positive training samples to compute the NCL losses.
>
> [1] Wang, Wenjie, et al. "Certified robustness to word substitution attack with differential privacy." Proceedings of the 2021 conference of the North American chapter of the association for computational linguistics: human language technologies. 2021.
>
> [2] Lecuyer, Mathias, et al. "Certified robustness to adversarial examples with differential privacy." 2019 IEEE symposium on security and privacy (SP). IEEE, 2019
>
> [3] Subramanian, Rishabh. "Have the cake and eat it too: Differential Privacy enables privacy and precise analytics." Journal of Big Data 10.1 (2023): 117.
>
> [4] Hemkumar, D., and Pvn Prashanth. "A Weighted Privacy Mechanism under Differential Privacy." 2023 4th International Conference on Intelligent Technologies (CONIT). IEEE, 2024.

---

### Official Review · Reviewer_MpcM · 2024-10-27

**Soundness:** 3
**Presentation:** 2
**Contribution:** 2
**Rating:** 6
**Confidence:** 5

**Summary:**

In this paper, the authors propose the first Non-Contrastive Learning (NCL) framework for Sequential Recommendation (SR). The authors design a novel preference-preserving profile generation method to produce high-quality positive samples for non-contrastive training. The paper is easy to follow and well-organized. The authors conduct extensive experiments to validate the efficacy of the proposed NCL-SR.

**Strengths:**

S1: The paper is easy to follow and well-organized.

S2: The authors provide theorem analysis for the proposed method which is novel and interesting to me.

S3: The authors conduct extensive experiments to validate the efficacy of the proposed NCL-SR.

**Weaknesses:**

W1: Some illustrations should be improved. For example, the authors should motivate why you choose Matrix Cross Entropy (MCE) against some other methods (e.g., Barlow Twin, MEC and DirectAU) for contrastive learning?

W2: Some method details are missing. For example, the calculation of $C(Z,Z’)$ in Eq.(9) should be provided to make the paper more readable.

W3: Since you need to figure out the eigenvalue of $\bm{V}$ in Eq.(9), how about the time complexity of your proposed method?

**Questions:**

Please refer to the Weaknesses above.

---

> ### Author Response · Authors · 2024-11-25
> **Response to Reviewer MpcM**
>
> We greatly appreciate Reviewer MpcM for the valuable feedback and suggestions. We are encouraged that the reviewer finds our paper well-organized, our method novel and our experiments extensive. To address the reviewer’s questions, we provide detailed clarifications and discussions below.
>
> **W1: Some illustrations should be improved. For example, the authors should motivate why you choose Matrix Cross Entropy (MCE) against some other methods (e.g., Barlow Twin, MEC and DirectAU) for contrastive learning?**
>
> We appreciate this valuable comment. To address the reviewer’s question, we clarify the following points:
>
> - The motivation why we choose MCE against some other methods is because of the generality of MCE. As shown in [1], Matrix Cross Entropy is a more generalized framework for NCL, and the MCE-based uniformity loss encompasses existing loss functions of several other NCL methods. Furthermore, we note that many NCL methods (including Barlow Twins and MEC) mainly focus on pursuing uniformity in representation learning. However, as shown in [2,3,4], alignment is also an important property to improve recommendation performance. We select Maxtrix Cross Entropy, because it can harmonize the alignment and uniformity with a unified framework. Specifically, based on Matrix Cross Entropy, there exist well-defined alignment loss and uniformity loss.
> - Finally, in terms of DirectAU, it directly pursues uniformity and alignment. However, it is a contrastive learning framework that computes training losses with in-batch contrastive samples, which makes it more computationally expensive.
> - We will add these clarifications in our final version.
>
>
> **W2: Some method details are missing. For example, the calculation of C(Z,Z′)  in Eq.(9) should be provided to make the paper more readable.**
>
> We apologize for the confusion. To address the reviewer’s question, we clarify the following point and will add it to our final version:
> - $C(Z,Z’)$ is the centered cross-correlation matrix between $Z$ and $Z’$. As in [1], it is calculated using the following equation: $C(Z,Z') = \frac{1}{B}Z H_B Z’^{T}$, with $H_B$ being the centering matrix: $H_B = I_B - \frac{1}{B} 1_B 1_B^T$.
>
>
> **W3: Since you need to figure out the eigenvalue of $V$ in Eq.(9), how about the time complexity of your proposed method?**
>
> We apologize for the confusion. We believe the reviewer refers to the $V$ in Equation (8), where we need to compute matrix-logarithm for $V$, i.e., $logV$. To address the reviewer’s question, we clarify that this term is approximated with Taylor Expansion as in [1]. Therefore, in general, the time complexity of this operation is the time complexity is $\mathcal{O}(md^2)$ (under our implementation), with $m$ being the order of the expansion and $d$ being the dimensionality of the matrix $V$. In our implementation, we set $m=4$ for computational efficiency and $d=768$ as given by the dimensionality of E5 outputs:
> $$
> \mathrm{logV} = \sum_{i=1}^{m} (-1)^{(m+1)}\frac{(V-I)^{m}}{m}.
> $$
>
> [1] Zhang, Yifan, et al. "Matrix information theory for self-supervised learning." arXiv preprint arXiv:2305.17326 (2023).
>
> [2] Wang, Chenyang, et al. "Towards representation alignment and uniformity in collaborative filtering." Proceedings of the 28th ACM SIGKDD conference on knowledge discovery and data mining. 2022.
>
> [3] Yang, Liangwei, et al. "Graph-based alignment and uniformity for recommendation." Proceedings of the 32nd ACM International Conference on Information and Knowledge Management. 2023.
>
> [4] Ou, Yangxun, et al. "Prototypical contrastive learning through alignment and uniformity for recommendation." arXiv preprint arXiv:2402.02079 (2024).

---

### Official Review · Reviewer_EEcc · 2024-10-31

**Soundness:** 3
**Presentation:** 2
**Contribution:** 3
**Rating:** 6
**Confidence:** 4

**Summary:**

The authors propose a Non-Contrastive Learning framework for Sequential Recommendation powered by preference preserving user profile generation: NCL-SR. The NCL-SR framework eliminates the computational overhead of identifying and generating negative samples in CL. The experimental results on various benchmark datasets and model architectures demonstrate the effectiveness of the NCL-SR method.

**Strengths:**

+ The methodology of this paper is technically sound. The method itself is somewhat novel to me.

+ To my best of knowledge, it is the first attempt to utilize matrix cross entropy in the recommendation system.

+ The proposed NCL-SR achieves much better performance against other baseline models.

**Weaknesses:**

+ Fig.1. needs to be polished. For instance, it is unclear for me what is the plot meaning of the exponential mechanism with polygons in Fig.1?

+ Why do you set the utility score as $\Delta_u = e - 1/e$? It is unclear for me. Could you please kindly provide some insights on that?

+ Could different values of $\gamma$ affect the model performance?

+ Some similar papers should be cited or discussed, e.g., [1] and [2].

Ref:

[1] DDGHM: Dual dynamic graph with hybrid metric training for cross-domain sequential recommendation

[2] Enhancing hierarchy-aware graph networks with deep dual clustering for session-based recommendation

**Questions:**

Please refer to the Weaknesses above.

---

> ### Author Response · Authors · 2024-11-25
> **Response to Reviewer EEcc (Part 1)**
>
> We greatly appreciate Reviewer EEcc for the valuable feedback and suggestions. We are encouraged that the reviewer finds our approach novel and technically sound. To address the reviewer’s questions, we provide detailed clarifications and discussions below.
>
> **W1: Fig.1. needs to be polished. For instance, it is unclear for me what is the plot meaning of the exponential mechanism with polygons in Fig.1?**
>
> To address the reviewer’s question, we will elaborate on polygons in Fig. 1 by adding the following details in the final version.
>
> - In terms of the first question, the three polygons represent three synonym sets for the first, second and fourth item in Fig. 1, respectively. Each synonym set is calculated with Equation 4. Regarding one polygon, the vertices within it represent the embeddings of the synonym items w.r.t. the original item from the user’s history. For instance, for the first item (i.e., the movie “Save the Private Ryan ”) of the user’s history, our method first encodes this item into an embedding vector. Then, we compute the synonym set for this item with Equation 4. Therefore, all vertices in the polygon represent the embedded items.  In addition to the vertices, there are dashed red lines in the polygons. The red dashed lines represent that the linked items are used for computing the exponential mechanism.
>
> - In terms of the second question, we clarify that there is no corresponding polygon for the third red text, because our approach only perturbs a part of the user profiles. In Fig. 1, the exponential mechanism is only applied for the first, second and fourth item. Such a design aims to reduce computation costs of the exponential mechanism.
>
> **W2: Why do you set the utility score as Δu=e−1/e? It is unclear for me. Could you please kindly provide some insights on that?**
>
> We appreciate this technical question. To address the reviewer’s confusion, we clarify that $\delta_u$ is defined as the sensitivity of the utility score. It is a constant value that is computed as the maximum utility score value minus minimum utility score value as in [1]. In the exponential mechanism, $\delta_u$ is a required factor to compute the sampling probability (Equation 6). In our design, the maximal utility score is $e^1$ and the minimum utility score is $e^-1$. Therefore, the sensitivity of the utility score is $e-\frac{1}{e}$. We will add these clarifications in our final version.
>
> [1] Mark Bun and Thomas Steinke. Concentrated differential privacy: simplifications, extensions, and lower bounds. In Theory of Cryptography Conference, 635–658. Springer, 2016.

---

> ### Author Response · Authors · 2024-11-25
> **Response to Reviewer EEcc (Part 2)**
>
> **W3: Could different values of γ affect the model performance?**
>
> We appreciate this question. To address the reviewer’s confusion, we will add the following clarification into our final version:
> - We acknowledge that different values of $\gamma$ could indeed affect the model performance. This is revealed in our sensitivity analysis w.r.t. $\lambda_2$ (Fig.2b). To see this, we further clarify that $\lambda_2$ is equivalent to a rescaled $\gamma$ in our implementation. As shown in [1], the combination of $L_{uniform} + L_{align}$ could be expanded as follows:
> $$
> L_{uniform}(Z, Z^{'}) + L_{align}(Z, Z^{'})
> $$
> $$
> = \mathrm{MCE}\big(\frac{1}{d}I_{d}, C(Z,Z^{'})\big) -\mathrm{tr}\big(C(Z, Z^{'})\big) + \gamma \cdot \mathrm{MCE}\big(C(Z,Z), C(Z^{'}, Z^{'})\big)
> $$
> $$= -\mathrm{tr}\big((\frac{1}{d}I_{d})\mathrm{log}(C(Z,Z^{'}))\big) - \gamma \cdot \mathrm{tr}\big(C(Z,Z)\mathrm{log}(C(Z^{'}, Z^{'}))\big) + \gamma \cdot \mathrm{tr}\big(C(Z^{'}, Z^{'})\big) + \mathrm{const.}
> $$
> - In the expansion in the third row, it is observed that both the **second term** and the **third term** are derived from **$L_{align}$**. More importantly, both these two terms are re-scaled by $\gamma$. Since the second term and third term correspond to $L_{align}$, we can instead use another scaling factor that directly adjusts the trade-off between $L_{uniform}$ and $L_{align}$. Specifically, we used $\lambda_1$ to re-scale $L_{uniform}$, and use $\lambda_2$ to re-scale $L_{align}$ with $\gamma$ being absorbed into $\lambda_2$.
> - Finally, we highlight that such an implementation (i.e., absorbing $\gamma$ into $\lambda_2$) also reduces the number of hyperparameters required for tuning.
>
> [1] Zhang, Yifan, et al. "Matrix information theory for self-supervised learning." arXiv preprint arXiv:2305.17326 (2023).
>
> **W4: Some similar papers should be cited or discussed, e.g., [1] and [2].**
>
> Thanks for recommending additional literature. After reading these two papers, we find that they are related to our work because both [1,2] involve contrastive learning. Therefore, we will cite these two papers and add more discussion about them in our related work section. In particular, we will add:
>
> - In [1], aiming at the  Cross-Domain Sequential Recommendation challenge, the authors propose a novel model via dual dynamic graph modeling and hybrid metric training with contrastive learning integration.
>
> - In [2], a novel hierarchy-aware dual clustering graph network (HADCG) model is designed to explore the inherent hierarchy structures from both item popularity and collaborations. WIth information regularizer for intra-session clustering and contrastive training for inter-session clustering,  HADCG achieves substantial performance improvements in session-based recommendation.
>
> [1] DDGHM: Dual dynamic graph with hybrid metric training for cross-domain sequential recommendation
>
> [2] Enhancing hierarchy-aware graph networks with deep dual clustering for session-based recommendation

---

### Official Review · Reviewer_u9dS · 2024-11-02

**Soundness:** 2
**Presentation:** 2
**Contribution:** 2
**Rating:** 6
**Confidence:** 5

**Summary:**

Summary:
This paper addresses the issue of high computational cost in contrastive learning (CL)-based sequential recommendation. Innovatively, it introduces non-contrastive learning into sequential recommendation. To tackle the representation collapse and alignment problems caused by solely relying on positive samples, a novel user preference-preserving profile generation method is proposed for the Non-Contrastive Learning framework for sequential recommendation (NCL-SR). NCL-SR is capable of learning generalized and robust user representations for sequential recommendation. Experimental results demonstrate that NCL-SR outperforms traditional SR models and CL-based SR models in terms of performance.

**Strengths:**

Strength：
1.This paper innovatively proposes a non-contrastive learning framework for sequential recommendation.
2.Inspired by Differential Privacy (DP), a theoretically guaranteed user preference-preserving data augmentation method is proposed to address representation collapse and preference inconsistency issues.
3.Two novel loss calculation methods are introduced.
4.Extensive experiments are conducted to verify the effectiveness, and the roles of feature alignment and uniformity in sequential recommendation are analyzed.

**Weaknesses:**

Limitation：
1.What does "s()" represent in Eq. 1? What does "f()" represent in Eq. 4?
2.What do the three polygons in the upper part of Fig. 1 represent? Why is there no corresponding polygon for the third red text?
3.During the construction of candidate user profiles, a item-level approach is adopted. How is the number of perturbations determined? Why weren't further experiments conducted to verify the effect of perturbation quantity on recommendation performance?
4.What is the split ratio of the training, validation, and test sets? Why wasn't it explicitly stated?
5.The motivation of this paper is to address the high computational cost and memory consumption in negative sample extraction of CL in SR. Why weren't further experiments conducted to verify the effect of NCL-SR on computational cost and memory consumption?
6.Further refinement of the text is needed. For instance, what is the relationship between “NCL SR” and “NCL-SR” in line 178? The term "differential privacy" in line 210 is in lowercase, while "Differential Privacy" in line 215 is in uppercase. Consistency should be maintained.

**Questions:**

Limitation：
1.What does "s()" represent in Eq. 1? What does "f()" represent in Eq. 4?
2.What do the three polygons in the upper part of Fig. 1 represent? Why is there no corresponding polygon for the third red text?
3.During the construction of candidate user profiles, a item-level approach is adopted. How is the number of perturbations determined? Why weren't further experiments conducted to verify the effect of perturbation quantity on recommendation performance?
4.What is the split ratio of the training, validation, and test sets? Why wasn't it explicitly stated?
5.The motivation of this paper is to address the high computational cost and memory consumption in negative sample extraction of CL in SR. Why weren't further experiments conducted to verify the effect of NCL-SR on computational cost and memory consumption?
6.Further refinement of the text is needed. For instance, what is the relationship between “NCL SR” and “NCL-SR” in line 178? The term "differential privacy" in line 210 is in lowercase, while "Differential Privacy" in line 215 is in uppercase. Consistency should be maintained.

---

> ### Author Response · Authors · 2024-11-25
> **Response to Reviewer u9dS (Part 1)**
>
> We greatly appreciate Reviewer u9dS for the valuable feedback and suggestions. We are encouraged that the reviewer finds our paper novel and our evaluation extensive. To address the reviewer’s questions, we provide detailed clarifications and discussions below.
>
> **W1: What does "s()" represent in Eq. 1? What does "f()" represent in Eq. 4?**
>
> We apologize for the lack of clarity. We will improve the clarity by adding the following definitions and explanations about s() and f() right after Eq. 1 and Eq. 4, respectively:
> - s() in Eq. 1 represents an arbitrary similarity scoring function in a canonical form. For instance, the scoring function used throughout this paper is the cosine similarity between the embedded user profiles and items, as defined in Equation 4.
> - f() in Eq. 4 represents the text-based recommender model used in our work. The text-based recommender model encodes user/item texts into latent representations as described in line 313.
>
> **W2: What do the three polygons in the upper part of Fig. 1 represent? Why is there no corresponding polygon for the third red text?**
>
> To address the reviewer’s question, we will elaborate on polygons in Fig. 1 by adding the following details in the final version.
>
> - In terms of the first question, the three polygons represent three synonym sets for the first, second and fourth item in Fig. 1, respectively. Each synonym set is calculated with Equation 4. Regarding one polygon, the vertices within it represent the embeddings of the synonym items w.r.t. the original item from the user’s history. For instance, for the first item (i.e., the movie “Save the Private Ryan ”) of the user’s history, our method first encodes this item into an embedding vector. Then, we compute the synonym set for this item with Equation 4. Therefore, all vertices in the polygon represent the embedded items.  In addition to the vertices, there are dashed red lines in the polygons. The red dashed lines represent that the linked items are used for computing the exponential mechanism.
> - In terms of the second question, we clarify that there is no corresponding polygon for the third red text, because our approach only perturbs a part of the user profiles. In Fig. 1, the exponential mechanism is only applied for the first, second and fourth item. Such a design aims to reduce computation costs of the exponential mechanism.

---

> ### Author Response · Authors · 2024-11-25
> **Response to Reviewer u9dS (Part 2)**
>
> **W3: During the construction of candidate user profiles, an item-level approach is adopted. How is the number of perturbations determined? Why weren't further experiments conducted to verify the effect of perturbation quantity on recommendation performance?**
>
> We appreciate this valuable comment. To address the reviewer’s comments, we will add the following additional experimental results and analysis in our final version:
>
> - As suggested by the reviewer, we conduct a sensitivity analysis w.r.t. the number of perturbations by changing it from 1 to 8. The results are reported in the tables below. From the tables, we observe that, in general, the model performance is similar when varying the number of perturbations from 1 to 4. Moreover, we observe that the performance starts degrading after the number of perturbations is larger than 5. This is expected, because, with too many perturbations, the initial model may generate wrong user preferences at the beginning. During training, such wrong user preferences are preserved, so the model is then trained to learn wrong user preferences, which degrades the performance.
>   - Beauty
>   | num. pert. | 1      | 2      | 3      | 4      | 5      | 6      | 7      | 8      |
> |------------|--------|--------|--------|--------|--------|--------|--------|--------|
> | R@10       | 0.0772 | 0.0779 | 0.0791 | 0.0742 | 0.0783 | 0.0749 | 0.0726 | 0.0747 |
> | N@10       | 0.0415 | 0.0427 | 0.0440 | 0.0396 | 0.0430 | 0.0408 | 0.0400 | 0.0404 |
> | R@20       | 0.1130 | 0.1139 | 0.1135 | 0.1098 | 0.1165 | 0.1141 | 0.1089 | 0.1113 |
> | N@20       | 0.0505 | 0.0517 | 0.0526 | 0.0486 | 0.0526 | 0.0507 | 0.0492 | 0.0496 |
>   - Games
>   | num. pert. | 1      | 2      | 3      | 4      | 5      | 6      | 7      | 8      |
> |------------|--------|--------|--------|--------|--------|--------|--------|--------|
> | R@10       | 0.1089 | 0.1123 | 0.1140 | 0.1101 | 0.1076 | 0.1107 | 0.1104 | 0.1085 |
> | N@10       | 0.0574 | 0.0600 | 0.0611 | 0.0572 | 0.0561 | 0.0582 | 0.0573 | 0.0583 |
> | R@20       | 0.1656 | 0.1664 | 0.1683 | 0.1650 | 0.1626 | 0.1647 | 0.1629 | 0.1607 |
> | N@20       | 0.0716 | 0.0736 | 0.0748 | 0.0710 | 0.0700 | 0.0717 | 0.0705 | 0.0714 |
>   - Sports
>   | num. pert. | 1      | 2      | 3      | 4      | 5      | 6      | 7      | 8      |
> |------------|--------|--------|--------|--------|--------|--------|--------|--------|
> | R@10       | 0.0397 | 0.0387 | 0.0441 | 0.0406 | 0.0382 | 0.0351 | 0.0358 | 0.0394 |
> | N@10       | 0.0210 | 0.0206 | 0.0237 | 0.0209 | 0.0201 | 0.0191 | 0.0192 | 0.0212 |
> | R@20       | 0.0593 | 0.0592 | 0.0660 | 0.0617 | 0.0596 | 0.0565 | 0.0549 | 0.0611 |
> | N@20       | 0.0259 | 0.0258 | 0.0292 | 0.0262 | 0.0255 | 0.0244 | 0.0240 | 0.0266 |
>   - Toys
>   | num. pert. | 1      | 2      | 3      | 4      | 5      | 6      | 7      | 8      |
> |------------|--------|--------|--------|--------|--------|--------|--------|--------|
> | R@10       | 0.0930 | 0.0958 | 0.0941 | 0.0942 | 0.0926 | 0.0908 | 0.0922 | 0.0928 |
> | N@10       | 0.0511 | 0.0540 | 0.0537 | 0.0540 | 0.0525 | 0.0515 | 0.0521 | 0.0522 |
> | R@20       | 0.1248 | 0.1309 | 0.1286 | 0.1314 | 0.1281 | 0.1238 | 0.1261 | 0.1302 |
> | N@20       | 0.0592 | 0.0628 | 0.0623 | 0.0634 | 0.0615 | 0.0598 | 0.0607 | 0.0615 |
>   - Office
>   | num. pert. | 1      | 2      | 3      | 4      | 5      | 6      | 7      | 8      |
> |------------|--------|--------|--------|--------|--------|--------|--------|--------|
> | R@10       | 0.0983 | 0.0989 | 0.1047 | 0.1028 | 0.0983 | 0.0875 | 0.0989 | 0.0980 |
> | N@10       | 0.0521 | 0.0524 | 0.0534 | 0.0529 | 0.0506 | 0.0431 | 0.0498 | 0.0509 |
> | R@20       | 0.1561 | 0.1535 | 0.1625 | 0.1570 | 0.1554 | 0.1459 | 0.1519 | 0.1551 |
> | N@20       | 0.0662 | 0.0662 | 0.0681 | 0.0665 | 0.0649 | 0.0579 | 0.0631 | 0.0653 |
>   - Auto
>   | num. pert. | 1      | 2      | 3      | 4      | 5      | 6      | 7      | 8      |
> |------------|--------|--------|--------|--------|--------|--------|--------|--------|
> | R@10       | 0.1390 | 0.1280 | 0.1354 | 0.1268 | 0.1220 | 0.1268 | 0.1256 | 0.1220 |
> | N@10       | 0.0690 | 0.0660 | 0.0714 | 0.0667 | 0.0630 | 0.0677 | 0.0672 | 0.0626 |
> | R@20       | 0.2207 | 0.2098 | 0.2085 | 0.2037 | 0.2049 | 0.1976 | 0.2024 | 0.1902 |
> | N@20       | 0.0894 | 0.0867 | 0.0899 | 0.0860 | 0.0839 | 0.0856 | 0.0863 | 0.0800 |
> - Finally, regarding the choice of 3 perturbations in our paper, we clarify that this choice is based on our empirical observations during experiments. Firstly, we searched for this hyperparameter based on the validation data of Beauty, and found that the configuration of 3 perturbations achieved best performance. Second, although the number of perturbations impacts the model performance, this hyperparameter is less influential than other hyperparameters that directly affect the training loss, namely $\lambda_1$ and $\lambda_2$. As such, we used 3 perturbations in our experiments.

---

> ### Author Response · Authors · 2024-11-25
> **Response to Reviewer u9dS (Part 3)**
>
> **W4: What is the split ratio of the training, validation, and test sets? Why wasn't it explicitly stated?**
>
> We apologize for missing this point. The split ratio of the training, validation, and test sets is 2:2:6 in our paper. Such a split ratio is selected because we aim to explore the model’s performance under a limited quantity of training data and the model’s generalization on cold-start users, following the recent studies [1,2,3,4]. For instance, in [4], there was only 10% training data used to train the model for the recommendation task. However, in our experiments, it is observed that many baselines could not converge under extreme data sparsity. Therefore, we increase the number of training samples for the baselines to converge (20%). This enables a fairer comparison between our method against the baselines. We will add these clarifications in our final version.
>
> [1] Wu, Xuansheng, et al. "Could Small Language Models Serve as Recommenders? Towards Data-centric Cold-start Recommendation." Proceedings of the ACM on Web Conference 2024. 2024.
>
> [2] Qian, Tieyun, et al. "Attribute Graph Neural Networks for Strict Cold Start Recommendation." 39th IEEE International Conference on Data Engineering, ICDE 2023. IEEE Computer Society, 2023.
>
> [3] Wang, Chunyang, et al. "Deep meta-learning in recommendation systems: A survey." arXiv preprint arXiv:2206.04415 (2022).
>
> [4] Large Language Models Make Sample-Efficient Recommender Systems
>
> **W5: The motivation of this paper is to address the high computational cost and memory consumption in negative sample extraction of CL in SR. Why weren't further experiments conducted to verify the effect of NCL-SR on computational cost and memory consumption?**
>
> We appreciate this valuable comment. To address the reviewer’s questions, we provide the following clarifications and additional experimental results as follows:
>
> - From a theoretical point of view, the time complexity of CL is $\mathcal{O}(n^2)$, if the training batch size is n and the loss is computed with in-batch negative samples. In comparison, the time complexity of NCL is $\mathcal{O}(n)$, because there are no negative samples at all. Therefore, we mainly cite some related work [1,2,3,4] to support the statement that NCL is more computationally efficient than CL.
>
> - Furthermore, we conducted additional experiments to evaluate the computational cost and memory consumption of computing CL and NCL losses. In this set of experiments, we ensure that all methods use exactly the same training configuration. Moreover, we vary the batch size from 1 to 5. The CUDA version we use is 12.1. The GPU used to evaluate memory consumption is NVIDIA A40 with 46 GB memory. We report the results in the table below. In the Table below, we report the relative memory consumption between NCL and CL $\frac{\text{memory consumption of NCL}}{\text{memory consumption of CL}}$:
>
> | Batch Size | 1    | 2    | 3    | 4    | 5                |
> |------------|------|------|------|------|------------------|
> | NCL/CL     | 0.72 | 0.57 | 0.51 | 0.51 | CL out of memory |
>
> - We will add the above results and analysis in our final version.
>
> [1] Jure Zbontar, Li Jing, Ishan Misra, Yann LeCun, and St´ephane Deny. Barlow twins: Self-supervised learning via redundancy reduction. In International conference on machine learning, pp. 12310– 12320. PMLR, 2021.
>
> [2] Jean-Bastien Grill, Florian Strub, Florent Altch´e, Corentin Tallec, Pierre Richemond, Elena Buchatskaya, Carl Doersch, Bernardo Avila Pires, Zhaohan Guo, Mohammad Gheshlaghi Azar, et al. Bootstrap your own latent-a new approach to self-supervised learning. Advances in neural information processing systems, 33:21271–21284, 2020.
>
> [3] Jaejin Cho, Jes´us Villalba, Laureano Moro-Velazquez, and Najim Dehak. Non-contrastive self- supervised learning for utterance-level information extraction from speech. IEEE Journal of Selected Topics in Signal Processing, 16(6):1284–1295, 2022.
>
> [4] Zhijian Zhuo, Yifei Wang, Jinwen Ma, and Yisen Wang. Towards a unified theoretical understanding of non-contrastive learning via rank differential mechanism. arXiv preprint arXiv:2303.02387, 2023.
>
>
> **W6: Further refinement of the text is needed. For instance, what is the relationship between “NCL SR” and “NCL-SR” in line 178? The term "differential privacy" in line 210 is in lowercase, while "Differential Privacy" in line 215 is in uppercase. Consistency should be maintained**
>
> We appreciate this comment. We will change “NCL SR” to “NCL-SR” in line 178 and throughout the paper. We will also change “differential privacy” in line 210 to “Differential Privacy” throughout the paper. We will check the consistency of the other terminology as well in the final version.

---

### Author Response · Authors · 2024-11-27
**General Response and Summary of Paper Updates**

We thank all reviewers for their valuable and insightful feedback. We are particularly grateful that the reviewers found our proposed approach novel (u9dS, EEcc, MpcM), and recognized our studied problem as important (A3iE). The reviewers also appreciated our theoretical analysis (u9dS, MpcM, A3iE), considered our work technically sound (EEcc), and commended the extensive experiments we conducted (u9dS, EEcc, MpcM, A3iE).

According to the suggestions and comments from the reviewers, we conduct a series of additional experiments and include the results in the revised paper. We also add more clarifications in our paper to address the questions raised by the reviewers.

Below is a summary of paper updates, corresponding to reviewers' comments.

- **Regarding the questions from Reviewer u9dS:**
    - **W1**: we add the definition of $s$ and $f$ at line 173 and line 247, respectively.
    - **W2**: we add the explanation for Fig 1 in Section 4.1 (line 295 - 302).
    - **W3**: we add the experiments w.r.t. the number of perturbations in Section 5.5 and Appendix E.
    - **W4**: we add the clarification about the data split ratio in Section 5.1
    - **W5**: we add the memory consumption evaluation of CL and NCL methods in Section 5.3 and Appendix G.
    - **W6**: we fix the typos in our paper and make the terminology consistent in line 178 and line 210.
- **Regarding the questions from Reviewer EEcc:**
    - **W1**: we add the explanation for Fig 1 in Section 4.1 (line 295 - 302).
    - **W2**: we add the clarification about the sensitivity score of the utility score at line 257.
    - **W3**: we add the clarification about the effect of $\gamma$ in Section 5.5 and Appendix E.
    - **W4**: we update Section 2 by citing the papers the reviewer recommended.
- **Regarding the question from Reviewer MpcM:**
    - **W1**: we add a motivation of choosing MCE over other NCL/CL methods in line 324 - 327.
    - **W2**: we add the equation for computing $C(Z, Z^{'})$ in Line 339.
    - **W3**: we add the clarification about computing $\log V$ in Section 4.2 and Appendix G.
- **Regarding the question from Reviewer A3iE:**
    - **W1**: we add the memory consumption evaluation of CL and NCL methods in Section 5.3 and Appendix G.
    - **W2**: we add the experiments for implementing SR models with E5 in Section 5.2 and Appendix F.
    - **Q1**: we add a detailed discussion between our method and DP in Section 4.1 and Appendix A.


We again greatly appreciate all reviewers' suggestions. We hope the updated results and our responses have addressed the questions that reviewers have. Please let us know if there are any other questions or suggestions.

Thanks,

Authors

---

### Meta-Review · Area_Chair_1iB2 · 2024-12-19

**Metareview:**

The paper titled "A Non-Contrastive Learning Framework for Sequential Recommendation with Preference-Preserving Profile Generation" introduces a novel approach to sequential recommendation (SR) by addressing the computational challenges inherent in traditional contrastive learning (CL) methods. The authors propose the first non-contrastive learning (NCL) framework tailored for SR, which eliminates the reliance on negative samples and thus reduces computational overhead. To mitigate the risk of representation collapse—common in the absence of negative samples—the authors design a preference-preserving profile generation method inspired by differential privacy. This method creates diverse and consistent user profiles that improve the alignment and uniformity of the learned representations. Experimental results across various benchmark datasets and architectures demonstrate the method's effectiveness, surpassing traditional CL-based SR methods and baseline SR models. The theoretical analysis of alignment and uniformity is another key contribution, with insights into their interplay in improving generalization.

The strengths of the paper are manifold. First, the work presents a clear innovation by adapting NCL to sequential recommendation, which, to my knowledge, has not been explored before. Second, the preference-preserving profile generation method offers both theoretical guarantees and practical utility, ensuring high-quality positive samples for non-contrastive training. Third, the paper's experimental rigor is commendable, with results showcasing significant improvements over existing methods. Additionally, the authors provide a thorough analysis of alignment and uniformity, which offers valuable insights for the research community. Finally, the reviewers highlighted the technical soundness and novelty of the proposed approach, further validating its scientific merit.

Despite its merits, the paper has some weaknesses. While the methodology is solid, the reviewers noted areas requiring clarification or refinement. For example, Reviewer u9dS pointed out inconsistencies in text presentation and the lack of clarity in defining certain terms and equations, such as "s()" and "f()". Reviewer EEcc requested a deeper explanation for specific design choices, like the utility score formulation and the plotting details in Figure 1. Reviewer MpcM raised concerns about the time complexity of eigenvalue calculations and the motivation behind choosing Matrix Cross Entropy (MCE) over other methods. Additionally, Reviewer A3iE suggested more detailed discussions on the connection between differential privacy and the proposed profile generation mechanism, as well as efficiency studies on computational cost and memory consumption.

The authors addressed these concerns effectively during the rebuttal period. They provided clarifications and new content in the revised paper, such as clearer definitions of key terms, additional experimental results on memory consumption and computational efficiency, and expanded discussions on related works and design decisions. The detailed response to Reviewer u9dS's questions, including new sensitivity analyses and performance evaluations, strengthens the paper's rigor. Similarly, the explanations offered to Reviewers EEcc, MpcM, and A3iE enhance the paper's presentation and address the raised points adequately.

The decision to recommend acceptance is based on several critical factors. First, the paper's contributions are both novel and impactful, addressing a significant limitation in sequential recommendation systems. The proposed method not only improves efficiency but also enhances representation learning quality, as demonstrated empirically and theoretically. Second, the authors' engagement during the rebuttal period and the substantial improvements made to the manuscript indicate a commitment to high-quality research. Lastly, while the reviewers raised some valid concerns, the authors addressed these comprehensively, leaving no significant gaps in the work.

**Additional Comments On Reviewer Discussion:**

The discussion among reviewers revolved primarily around the technical clarity and implementation details of the proposed method. Reviewer u9dS raised questions about the definition of certain functions, the interpretation of plots in Figure 1, and the split ratios for training and test data. These were clarified in the rebuttal with additional explanations and updates to the paper, including sensitivity analyses and memory consumption evaluations. Reviewer EEcc sought insights into the utility score formulation and recommended citing additional related works. These issues were addressed with an expanded discussion and proper citations in the final version. Reviewer MpcM highlighted the choice of Matrix Cross Entropy and computational complexity concerns, which the authors justified convincingly by emphasizing the framework's generality and providing approximations for computationally intensive steps. Finally, Reviewer A3iE requested further justification for the use of differential privacy and additional comparisons to ensure fairness. These points were also resolved, with the authors providing detailed theoretical and empirical evidence to support their claims.

In weighing these points, the reviewers' initial ratings, and the authors' responses, I concluded that the authors effectively addressed all major concerns. The added experimental and theoretical contributions during the rebuttal period further strengthen the paper. Overall, this paper represents a valuable contribution to the field of sequential recommendation, and I recommend its acceptance.

---

### Decision · Program_Chairs · 2025-01-22

Accept (Poster)